# Exposure of Cattle Breeding Herds to Naturally Co-Contaminated Zearalenone and Deoxynivalenol: The Relevance of a Urinary Mycotoxin Monitoring System for Herd Health and Food Safety

**DOI:** 10.3390/toxins16090402

**Published:** 2024-09-18

**Authors:** Oky Setyo Widodo, Seiichi Uno, Emiko Kokushi, Osamu Yamato, M. Fariz Fadillah Mardianto, Urara Shinya, Yuto Kano, Chiho Kawashima, Yasuo Fushimi, Tetsushi Ono, Masayasu Taniguchi, Mitsuhiro Takagi

**Affiliations:** 1Joint Graduate School of Veterinary Sciences, Yamaguchi University, Yamaguchi 753-8515, Japan; oky.widodo@fkh.unair.ac.id (O.S.W.); masa0810@yamaguchi-u.ac.jp (M.T.); 2Division of Animal Husbandry, Faculty of Veterinary Medicine, Airlangga University, Surabaya 60115, Indonesia; osam@vet.kagoshima-u.ac.jp; 3Faculty of Fisheries, Kagoshima University, Kagoshima 890-0056, Japan; uno@fish.kagoshima-u.ac.jp (S.U.); kokushi@fish.kagoshima-u.ac.jp (E.K.); 4Joint Faculty of Veterinary Medicine, Kagoshima University, Kagoshima 890-0065, Japan; 5Department of Mathematics, Faculty of Science and Technology, Airlangga University, Surabaya 60115, Indonesia; m.fariz.fadillah.m@fst.unair.ac.id; 6Soo Agricultural Mutual Aid Association, Kagoshima 899-8212, Japan; shinya-u@nosai46.jp (U.S.); kano-y@nosai46.jp (Y.K.); 7Field Center of Animal Science, Obihiro University of Agriculture and Veterinary Medicine, Obihiro 080-8555, Japan; kawasima@obihiro.ac.jp; 8Guardian Co., Ltd., Kagoshima 890-0033, Japan; yasuo243@guardian-vet.com; 9Joint Faculty of Veterinary Medicine, Yamaguchi University, Yamaguchi 753-8515, Japan; yt-ono@yamaguchi-u.ac.jp

**Keywords:** zearalenone, deoxynivalenol, anti-Müllerian hormone, serum amyloid A, 8-hydroxy-2′-deoxyguanosine, breeding cow, food supply chain

## Abstract

The widespread presence of Fusarium mycotoxins in animal feed is a global issue, not only for the health of livestock but also for ensure the safety of food as an end product. High concentrations of zearalenone (ZEN) and deoxynivalenol (DON) have been detected in the diets of Japanese Black (JB) and Holstein Friesian (HF) breeding herds. Consequently, we monitored serum biochemical parameters over a long time in both herds, focusing on anti-Müllerian hormone (AMH) levels and acute-phase inflammation. Additionally, urinary 8-hydroxy-2′-deoxyguanosine (8-OHdG) and progesterone levels were measured in the HF herd. The JB herd, a ZEN-dominant model with low DON contamination, demonstrated ZEN levels that exceeded the Japanese limit in the purchased total mixed rations (TMR). Conversely, the HF herd, which primary consumes DON-dominant feed with low ZEN contamination, had high DON levels in the dent corn silage. Specifically, the JB herd’s TMR contained 1.79 mg/kg ZEN and 0.58 mg/kg DON, whereas the HF herd’s silage had 15.3 mg/kg DON (dried sample) and 0.1 mg/kg ZEN. Enzyme-linked immunoassay were used to measure urinary ZEN-DON levels following confirmation through liquid chromatography-tandem mass spectrometry. Urinary ZEN-DON levels measured were significantly correlated (*p* < 0.05, *r* > 0.6) in both herds. In the HF herd, AMH levels increased (*p* = 0.01) and serum amyloid A (SAA) levels decreased (*p* = 0.02) when contaminated and at the end of the monitoring period. Additionally, urinary ZEN and DON levels were significantly correlated with SAA levels (ZEN: *p* = 0.00, *r* = 0.46; DON: *p* = 0.03, *r* = 0.33), with an increase in ZEN and DON levels resulting in higher SAA levels. The JB herd showed no significant differences. Additionally, in the HF herd, 8-OHdG/Cre levels increased significantly during major contamination periods (*p* < 0.05). Clinical data from the HF herd indicated an increase in mastitis cases and treatment rates during periods of major contamination. Abortion rates in the HF herd decreased from 22.9% (before monitoring) to 8.9% (during the high contamination period) and finally to 1% (at the end of the monitoring period), with corresponding increases in progesterone levels. ZEN-DON contamination adversely affects breeding cattle’s productivity, reproductive performance, and health. Therefore, monitoring urinary ZEN-DON is valuable for detecting contaminants and ensuring the safety of food products.

## 1. Introduction

Mycotoxins are secondary metabolites produced by fungi that contaminate cattle feed at various stages, including the field growth, harvest, transportation, feed manufacturing, and storage [1,2,3]. The widespread presence of Fusarium mycotoxins in animal feed is a global issue for the health of livestock and for the safety of food as an end product. We monitored beef and dairy cattle herds to examine the dynamics of mycotoxin contamination at meat and milk production sites, which are major sources of human food. Furthermore, food safety depends not only on mycotoxin contamination of livestock products but also on the health of cattle at the source of contamination. Previously, researchers have studied its toxicity, its effect on organisms, and methods to control these compounds. Zearalenone (ZEN) is an estrogenic mycotoxin that interferes with cattle reproductive functions and is frequently found in grains and feed [4,5,6]. We previously developed and validated a urinary ZEN monitoring system for cattle using an enzyme-linked immunoassay (ELISA) and liquid chromatography-tandem mass spectrometry (LC/MS/MS). This study aimed to investigate production sites and determine the impact of mycotoxin contamination on cattle health and fertility [7,8,9]. In addition, we evaluated the efficacy of ZEN adsorbents added to feed in reducing mycotoxin absorption using a urine ZEN monitoring system [7]. Our recent study [10] explored the effects of ZEN contamination on two biomarkers: anti-Müllerian hormone (AMH), a potential marker of fertility, ovulatory hyper-response, ovarian dysfunction, and herd longevity [11,12,13,14], and serum amyloid A (SAA), an acute-phase protein associated with inflammation [15,16,17,18].

Several types of fungi produce multiple toxins. The two main mycotoxins produced by Fusarium species are ZEN and deoxynivalenol (DON). DON is cytotoxic and immunotoxic and can impair protein synthesis. Susceptibility to DON varies among animals, with ruminants being fairly resistant and pigs being particularly vulnerable [19]. Co-contamination with ZEN and DON is common, occurring in approximately 23.8% of 400 feedstuff samples tested and 27% of cattle feed samples evaluated [20]. Improved detection tools have enabled the identification of multiple mycotoxins in feed, aiding the development of management strategies to minimize their adverse effects on cattle health and reproductive performance.

Exposure tests using in vitro cultured cells are important for evaluating toxicity at the genetic level [21,22,23]. To elucidate the effects of mycotoxin contamination on cattle health and fertility, it is important to investigate naturally occurring cases. In our ongoing investigation of ZEN infiltration in cow herds, we confirmed ZEN and DON contamination in the feed of two breeding herds: Japanese Black (JB) and Holstein Friesian (HF). This study used these herds as models for ZEN- and DON-dominant contamination to evaluate its effects on health and fertility. We verified the efficacy of the DON measurement system in urine samples using a commercially available ELISA kit. Additionally, we conducted blood biochemical tests, measuring levels of progesterone (P4) and urinary 8-hydroxy-2’-deoxyguanosine (8-OHdG), a biomarker of DNA oxidative stress [24,25]. Furthermore, we verified the relationship between changes in conception rate (CCR), abortion rate, disease incidence, and mycotoxin concentrations in urine and feed, based on the clinical records of both herds.

## 2. Results

### 2.1. Co-Contamination Status in Cattle Breeding Herds

Mycotoxin analysis of feed from JB and HF breeding herds (Table 1) revealed contamination with ZEN and DON. ZEN contamination was dominant in the JB herd, with the purchased total mixed rations (TMR) showing a peak level of 1.791 mg/kg, exceeding Japan’s limit of 1 mg/kg. In contrast, the HF herd’s feed was predominantly contaminated with DON, with corn silage containing 6.6 mg/kg, surpassing Japan’s 4 mg/kg limit for calves older than 3 months.

An ELISA detected ZEN and DON in the urine of both herds. In the JB herd, the ZEN levels peaked in July 2022 and declined until December 2022 (*p* = 0.06), whereas DON levels showed no significant differences across the evaluated time period (*p* = 0.55). The HF herd exhibited a peak ZEN contamination in November 2022, with a significant decrease recorded thereafter (*p* = 0.00). DON levels in the HF herd were the highest in August and September 2022, followed by a significant decrease (*p* = 0.00). The ZEN levels in the HF herd were significantly higher from August–November 2022 than in March and August 2023 (*p* < 0.05, except for September 2022 vs. March 2023, *p* = 0.06).

LC-MS/MS validation of urine samples (Table 2) showed significant correlations with ELISA results for ZEN in both herds (JB: *p* = 0.00, r = 0.94; HF: *p* = 0.00, r = 0.62) and for DON in the HF herd (*p* = 0.00, r = 0.75). In the HF herd, significant differences in ZEN and DON levels were observed across sampling periods (both *p* = 0.00), whereas the JB herd showed a tendency toward significant differences in ZEN levels (*p* = 0.055). The post hoc analysis results for the HF herd are presented in Appendix A. Figure 1 illustrates the urinary co-contamination levels in both herds, as determined by ELISA and LC-MS/MS.

### 2.2. Biomarker Analysis

We examined multiple biomarkers including AMH, SAA, and blood biochemical parameters in both herds and 8-OHdG and P4 in the HF herd. In the JB herd, AMH levels showed no significant differences across the periods (*p* = 0.67), whereas SAA levels showed a significant trend (*p* = 0.05). AMH levels (mean ± SEM) from July to December 2022 were 945.89 ± 158.82, 880 ± 112.64, 1012 ± 227.05, 522.5 ± 89.86, and 888.8 ± 171.52, respectively, indicating a decline. SAA levels for the same months were 7.24 ± 2.36, 6.97 ± 2.73, 2.51 ± 1.21, 2.44 ± 0.48, and 10.52 ± 7.06.

In the HF herd, AMH levels (mean ± SEM) were 447.2 ± 53.91 in August 2022, decreasing to 412.2 ± 53.82 in September and 380.22 ± 54.99 in November, and then increasing to 450.70 ± 48.44 in March 2023 and peaking at 529.85 ± 65.97 in August 2023 (*p* = 0.42). SAA levels were 5.79 ± 0.81 in August 2022, peaking at 15.01 ± 5.67 in September, then decreasing to 12.03 ± 5.23 in November and 3.45 ± 0.92 in March 2023, with a small increase to 4.80 ± 1.72 in August 2023 (*p* = 0.14).

Comparing AMH and SAA levels from August 2022 to August 2023 showed significant differences (ps of 0.01 and 0.02, respectively), with AMH levels increasing and SAA levels decreasing. Both ZEN and DON were positively correlated with SAA in the HF herd (ZEN: *p* = 0.00, r = 0.46; DON: *p* = 0.03, r = 0.33), with increases in ZEN and DON levels corresponding to increases in SAA levels. Figure 2 shows the AMH and SAA levels in both herds, as well as changes in SAA and urinary ZEN and DON levels.

Blood biochemical analyses were performed to measure the levels of GOT, GGT, FFA, T-Cho, BUN, Glu, Ca, IP, Mg, TG, Vit. A, Vit. E, 3HB, TP, Alb, and AG in the JB and HF breeding herds. In the JB herd, significant differences (*p* < 0.05) were observed across the evaluated time period for almost all parameters, except GOT (*p* = 0.06), Ca (*p* = 0.07), GGT (*p* = 0.45), Mg (*p* = 0.88), and Alb (*p* = 0.72). In contrast, in the HF herd, significant differences (*p* < 0.05) were found for all parameters except for Vit. E (*p* = 0.24) and TP (*p* = 0.39). Results of blood biochemical analyses (mean ± SEM) of the JB and HF breeding herd are presented in Appendix A, as Appendix A. The mean patterns of blood biochemical analyses for both herds are shown in Figure 3.

In the HF breeding herd, 8-OHdG/Cre levels were examined in eight cows at five time points from August 2022 to August 2023. Mean values (±SEM) ranged from 39.17 ± 2.12 to 76.21 ± 14.24, excluding one cow with extremely high levels. Statistical analysis revealed significant differences across sampling intervals (*p* = 0.01), with significant increases between August and November 2022 (*p* = 0.01), and higher levels in September and November 2022 than in August 2023 (*p* = 0.01 and *p* = 0.00, respectively). Figure 4a,b shows the mean 8-OHdG/Cre levels in the HF breeding herd and non-contaminated control herd, which showed no significant differences (*p* = 0.99). The P4 levels in the HF herd evaluated in August, September, and November 2022 showed statistically significant differences (*p* < 0.05), with a significant increase from August to September and in November (*p* < 0.05). Figure 4c presents the P4 levels (mean ± SEM) for each sampling period in the HF breeding herd.

### 2.3. Clinical Information

Clinical data for the JB and HF breeding herds are shown in Figure 5. For herd JB (Figure 5a), CCRs were monitored from January 2022 to March 2023, with the lowest CCR reported in June 2022 (18.75%) and the highest in October 2022 (87.5%). In response to contamination, a mycotoxin adsorbent was added to the feed from mid-July to late November. Three miscarriages occurred in cows inseminated between November–December 2022 and February 2023 at 7–8 months of gestation. For the HF herd, clinical information was recorded from March 2022 to March 2023, and mycotoxin contamination was verified from July to November 2022. The CCRs before, during, and after contamination were 39.82%, 37.84%, and 35.77%, respectively, with the lowest CCR (14.3%) being reported in September. The highest abortion rate (22.92%) occurred before contamination and peaked in April (28.8%). Figure 5b shows the CCR and abortion rates for the HF herds. Figure 5c shows the somatic cell count (SCC), incidence of mastitis, and treatment rates. SCC levels showed an increasing trend. The levels before, during, and after contamination were 6.17 × 10^4^, 7.92 × 10^4^, and 8.35 × 10^4^, respectively. The incidence of mastitis was the highest during contamination (17.5%), decreasing to 13.97% post-contamination. The treatment rate peaked at 4.72% during contamination, compared to 2.2% pre-contamination and 2.57% post-contamination.

## 3. Discussion

Global warming has increased mycotoxin contamination in cattle feed, prompting extensive research on three key components. The first focuses on the characteristics, toxicity, and metabolites of mycotoxins in vitro and in vivo [26,27]. The second examines the presence of mycotoxins, often found with other fungi, in animal food and body fluids such as serum, urine, and milk [28,29,30,31]. The third area encompasses feed management, fungal infection control, and physical, chemical, enzymatic, and biological degradation methods to reduce harmful effects on animals [32,33]. Ensilaged grass or hay may contain mycotoxins from pre-harvest contamination by *Fusarium* spp. and post-harvest contamination by fungal species commonly found in silage [34]. The co-contamination of feed with ZEN and DON, including the metabolites, is common in farms [6,20,35]. The clinical signs of mycotoxin exposure vary depending on the trichothecene type, dose, species, route, and exposure type [36]. Treatments are limited to symptomatic and supportive care with prevention focused on avoiding exposure [36]. A comprehensive database is crucial for understanding the effect of multi-mycotoxin contamination on reproduction and productivity and for monitoring remediation responses. This study investigated the effects of ZEN and DON co-contamination on cattle health and productivity, using herds naturally affected by these contaminants as models. The JB herd represented the ZEN-dominant model, whereas the HF herd represented the DON-dominant model. Continuous monitoring revealed significant effects on health, metabolism, and reproductive capabilities, with DON-dominated contamination significantly impairing liver function that persisted even after remediation.

We successfully identified two contaminated herds during the hot season in July, when cattle are more susceptible to mycotoxin contamination. Although the herds were affected by different mycotoxins, this presented an opportunity for a novel field study. The veterinarian for JB herds reported decreased monthly pregnancy rates and improper corpus luteum development during pregnancy diagnoses, prompting urinary ZEN monitoring. Our study confirmed ZEN contamination with DON as a sub-contaminant in the feed (TMR). Simultaneous contamination of *Fusarium* spp. with ZEN and DON prompted us to investigate urinary DON measurement systems. We used a commercially available ELISA kit to measure DON contamination, which was similar to our established urinary ZEN monitoring system [7,9]. Previous studies have reported the concurrent presence of ZEN and DON in animal feeds [6,20,35]. Therefore, we developed a system to measure DON in urine samples in conjunction with our urinary ZEN monitoring method. The HF herd served as a model for identifying DON-dominant contaminants in feed.

The HF herd manager reported increased cases of clinical mastitis, reduced milk output, more abortions, lower reproductive viability, and odd feed smells, leading to the suspicion of DON contamination. The herd primarily consumed homegrown dent corn silage, and clinical symptoms in similar herds during warmer weather suggested mycotoxin exposure. Two laboratories measured feed mycotoxin levels and found significant differences in the results (Table 1). Mycotoxin concentrations vary according to the collection location [37,38]. If only one laboratory reported DON levels below the contamination limit, DON contamination might not be suspected as the cause of mastitis or reduced fertility.

DON exposure is assessed through the analysis of urine and feces, with approximately 20% of ingested DON being recovered as 96% deepoxy-deoxynivalenol (DOM-1) and 4% DON [39]. Antibody-antigen immunoassays are ideal for routine mycotoxin detection [36], and routine immunoassessments are necessary for effective cattle risk management [10,36]. Given the time and cost associated with measuring DON, we used commercially available ELISA kits to detect DON contamination in cattle herds. While immunochemical chromatography of DON is simple and affordable, its cross-reactivity and matrix effects must be explored. Advances in biomarker research have enabled the determination of DON and its metabolites in urine using single or multiple indicators [36]. DON and its acetylated derivatives 3-acetyldeoxynivalenol (3-Ac-DON) and 15-acetyldeoxynivalenol (15-Ac-DON) are produced by plants. In mice, 3-Ac-DON exposure causes oxidative damage, cell death, immune cell infiltration, and impaired liver function [40]. We used ELISA kits with antibodies that cross-reacted with both DON and 3-Ac-DON, showing cross-reactivity rates exceeding 100%. Studies reported cross-reactivity levels of 520 ± 110 for 3-Ac-DON and 80 ± 10 for deoxynivalenol-3-glucoside (DON-3G) [41]. Significant DON metabolite contamination in the tested samples indicated that the results reflected total toxin levels, not just DON levels. Thus, the kit served as a group detection tool for DON, 3-Ac-DON, and DON-3G, rather than solely for DON [41]. Recent reports on 3-Ac-DON toxicity, in conjunction with DON, suggest that monitoring both DON and 3-Ac-DON as “total DON” in bovine urine is beneficial. Based on preliminary recovery tests, with or without β-glucuronidase incubation, we recommend using ELISA kits to screen urinary DON concentrations. These results should be validated with LC-MS/MS, and ongoing monitoring of the DON-contaminated herd is essential to establish a comprehensive DON measurement system.

One goal was to validate our ZEN and DON detection systems using ELISA and LC-MS/MS methods. These results indicated a significant need for a biological material measurement system, particularly for urine samples. Figure 1 shows a robust and statistically significant relationship between the ELISA and LC-MS/MS results for the two herds. Mycotoxin contamination of feed within herds is critical for herd management. Herd managers should consider the potentially harmful effects of mycotoxin contamination on the health and products of cattle. In the worst-case scenario, all contaminated feed must be disposed. Prompt identification of potential mycotoxin contaminants, assessment of cleanup efficacy, and ongoing herd monitoring are essential. Thus, we conveniently evaluated ZEN and DON contamination levels using an ELISA system to measure DON in urine, alongside a previously established ZEN concentration system. Both urinary ZEN and DON contamination were measured simultaneously. DON levels were measured during the first 120 min of ZEN incubation. This assessment included determination of the concentration and contamination ratios of each mycotoxin, as shown schematically in Figure 6. This is the first practical validation of urinary ZEN and DON monitoring systems in dairy and beef herds.

The greatest advantage of using urine samples to monitor the herd mycotoxin contamination status, as previously reported [9,10], is the ability to monitor and compare the concentrations actually ingested and absorbed from the intestinal tract. The intake volume of contaminated feed may affect the urinary mycotoxin concentration, making it a suitable method for monitoring and comparing mycotoxin exposure in cattle whose daily feed amount remains constant across each herd. To investigate the effects of chronic mycotoxin exposure on the health status and productivity of livestock herds, a urinary mycotoxin monitoring system is required to monitor mycotoxin intake from dietary feed [9]. This study revealed that simultaneous screening for ZEN-DON, which is frequently reported, is a crucial strategy to assess mycotoxin contamination in cattle herds. ELISA and LC-MS/MS concentration measurements suggested that ELISA-based urinary ZEN-DON assays may be accurate, simple, and effective for evaluating mycotoxin penetration kinetics and long-term chronic exposure at low doses.

The purpose of this study was to clarify the effects of mycotoxin exposure using serum biochemical analysis to monitor the general metabolic status after subsequent remediation. The DON-dominant herd (HF herd) exhibited significant changes in various biochemical parameters as the urinary DON concentration decreased, indicating improvements in liver function and nutritional status. However, the monitoring period showed significant changes in almost all the parameters, albeit within normal physiological ranges. GOT and GGT, indicators of liver function, significantly decreased after the counter-measures but remained elevated above normal ranges. We observed significant differences in some parameters in the ZEN-dominant (JB herd) group but did not observe any significant differences in indicators of liver function and nutritional status. Therefore, our results clearly indicated that the adverse effects on liver function were severe in DON-contaminated cattle, and in ZEN-contaminated cattle, as reported previously [37], there appeared to be no effect on liver function (GOT or GGT). Although there are few reports on the effects of ZEN-DON and its metabolites on hematological and serum biochemical parameters in cattle under farm conditions, the effects of Fusarium toxin intake (mainly ZEN and DON) on the serum biochemical parameters of protein metabolism and liver damage in cows have been reported [42]. Fusarium toxin contamination of dietary feed (0.09 mg ZEN/kg DM [dry matter] and 8.21 mg DON/kg DM) was insufficient to induce toxicological changes in serum hematological variables such as GOT, GGT, TP, and Alb. Clinical manifestations vary depending on the specific trichothecene, dosage, method of exposure, and type of exposure. Compared to pure toxins, spontaneous or experimental exposure to field-contaminated materials may yield different effects [36]. More databases are needed to determine how ZEN-DON co-contamination affects the biochemical profiles of cow herd blood.

Urinary 8-OHdG concentrations were higher in cattle predominantly contaminated with DON (herd HF). However, the concentration in the non-contaminated herd (control herd) did not change significantly, indicating that heat stress was not the cause of this increase (Figure 7 shows monthly temperatures). We hypothesize that oxidative damage caused DNA damage during DON contamination. Moreover, our findings suggest that heat stress has no impact on the 8-OHdG assay and that there are differences in the concentration of this biomarker among various cow breeds. To the best of our knowledge, no previous studies have reported the levels of 8-OHdG in the urine of mycotoxin-contaminated herds. In addition, we found that DON-dominated contamination caused oxidative DNA damage in cattle. Further investigations and data collection are necessary to gain a comprehensive understanding of the oxidative damage induced by mycotoxins in animals. Interestingly, in the HF herd, one cow had an extremely high urinary 8-OHdG concentration during the experimental period. Our observation of changes in the clinical manifestations in this cow may be an intriguing attempt to clarify the clinical utility of 8-OHdG concentration.

Recent research on in vitro cultures has provided detailed insights into the effects of DON and its metabolites on cow follicular development and herd fertility [43,44]. We hypothesized that certain effects on cow reproduction could be identified by evaluating the levels of mycotoxins and reproductive performance in the two contaminated herds. To verify this, we analyzed the reproductive performance (monthly conception rate) of the two herds over a period of 4–6 months before and after the implementation of the mycotoxin contamination management strategy. Figure 5 shows that our monitoring system detected mycotoxin contamination in the ZEN-dominant JB herd shortly after the conception rate decreased. In the DON-dominant HF herd, the monthly abortion rate remained high (20–30%) for several months, but decreased early when mycotoxin contamination was detected. From October to November 2022, we observed an increase in the abortion rate in the HF herd, potentially linked to increasing urinary ZEN concentrations, despite a decrease in urinary DON concentrations. Clinical results from these model herds revealed that elevated feed mycotoxin concentrations can significantly affect herd fertility rates, and that early contamination control measures can help herds recover faster.

In the ovary, AMH inhibits the recruitment of primordial follicles into the pool of developing follicles and decreases ovarian FSH responsiveness [45,46]. AMH is the most reliable endocrine marker of small antral gonadotropin-responsive follicles [45]. Several studies have associated ZEN and its metabolites with granulosa cell death and follicular atresia [33,47,48,49]. We found that natural ZEN exposure may influence AMH concentration and antral follicle count [37]. Additionally, our previous study revealed significant correlation between urinary ZEN and blood AMH and SAA concentrations in a JB breeding herd, determined based on monthly monitoring over a 2-year period. Our findings revealed a negative correlation between urinary ZEN concentrations two months earlier and AMH concentrations in the current month [10]. Despite the lack of significant differences in AMH concentrations for each period in the ZEN-dominant JB herd, AMH kinetics were the same as in our earlier results when ZEN contamination occurred. Similarly, in the case of the DON-dominant HF herd, the AMH levels in August 2022 decreased during the two subsequent sampling periods, in September and November 2022. Subsequently, AMH levels increased again in March 2023 and peaked in August 2023. The kinetics of AMH were similar to those of ZEN. On the other hand, Figure 2 illustrates the relationship between ZEN and SAA and DON and SAA. This shows that an increase in ZEN and DON levels leads to a subsequent increase in SAA levels, and vice versa.

However, we observed an interesting negative regression trend between ZEN and SAA concentrations in the ZEN-dominant herd (JB herd). This case showed a different trend of SAA concentrations compared to the DON-dominant herd (HF herd), with no increase in SAA concentration during feed contamination, but a decrease after contamination measures, as shown in Figure 2b. We previously reported an association between ZEN and SAA during the monitoring of ZEN-contaminated herds [9]. Reports indicate that ZEN not only disrupts the endocrine system but also causes immunotoxicity [50,51]. Other studies have shown that ZEN exposure alters the immune response of liver cells and inhibits the release of IL-1, IL-6, and TNF-α [50,51,52]. Therefore, the negative regression trend between urinary ZEN and SAA concentrations observed in this study supports our previous report and may be attributable to innate immunosuppression in cattle resulting from acute ZEN exposure. In the future, it will be necessary to increase the number of cow herds monitored, broaden the extent of monitoring, and clarify whether improving feed while detecting naturally exposed herds facilitates an improvement in productivity.

We previously reported that ZEN concentrations exceeding 1 mg/kg in the culture medium may reduce the meiotic competence of bovine oocytes, although it does not affect fertilization or subsequent development after in vitro fertilization [53]. Natural-feed ZEN contamination below the highest level allowed by Japanese law (<1 mg/kg) did not affect the development of superovulating embryos in JB and HF cows [54]. ZEN-contaminated feed affected cattle fertility by affecting uterine embryonic development after implantation. In the present study, the average P4 levels in the DON-dominant HF herd (30 cows) were significantly different (*p* < 0.05) during August, September, and November 2022 (Figure 4c), with a significant increase in P4 levels (*p* < 0.05) from August to September and November. This study did not include information on the cows’ estrous cycles or pregnancies. Administering 250 mg of purified ZEN orally to heifers for three estrous cycles led to a conception rate of 62%, compared to 87% in the control heifers, and oral administration of 500 mg of ZEN for two estrous cycles resulted in normal serum P4 levels [55,56]. As mentioned previously, spontaneous or experimental exposure to field-contaminated materials may provide results different from those of purified toxins [36]. Apoptosis in granulosa cells of ZEN- and DON-contaminated herds [21,33] suggests that luteinogenesis defects induce a decrease in P4 concentration during the peak of ZEN and DON contamination, which in turn leads to an increase in abortions. The results supported the presence of luteal dysplasia, as noted by the managing veterinarian during the high incidence of abortions in the ZEN-dominant (JB) herd. This is the first study to address this issue; however, more field investigations are required to understand how mycotoxins affect the luteinizing potential, P4 concentrations after follicular ovulation, and pregnancy in mycotoxin-contaminated herds.

Clinical records have demonstrated that ZEN-DON contamination affects not only fertility, but also the occurrence of mastitis. In the present study, farm managers of the DON-dominant HF herd observed an increasing clinical trend of mastitis. Exposure to grass silage containing high amounts of DON causes toxic syndrome in cattle, characterized by an increase in inflammatory reactions in the form of mastitis and laminitis [34]. Furthermore, milk yield decreases and somatic cell counts increase following exposure to moldy silage containing measurable amounts of ZEN and DON [34]. DON has been reported to inhibit lymphocyte proliferation in cattle, with the greatest decrease observed in bovine peripheral blood mononuclear cells (PBMC). It was speculated that DON might also pose a risk to the immune systems of animals typically considered to possess low susceptibility to DON, such as ruminants. This effect suggests that under certain conditions, such as inflammation or gastric diseases, low dietary DON concentrations could potentially impair animal heart health by acting as an immunosuppressive agent [22]. Recent in vitro research has shown that DON affects the proliferation and functional differentiation of bovine mammary alveolar cells, possibly by disrupting tight junctions and causing morphological alterations [57]. Figure 5c shows that the incidence of mastitis was 17.5% during periods of DON contamination; however, after implementing measures to address the contaminated feed, it decreased to 14.0% and further dropped to a low of 4.1%. However, the SCC increased continuously during the monitoring period, indicating that mastitis caused by DON-dominant contamination may have a prolonged effect on the SCC in dairy herds. Even at low levels, co-contamination can pose a health problem because of the possible additivity, antagonism, or synergy of their effects [20,58]. Given its significant effect on dairy herd productivity, further research in this area is necessary.

## 4. Conclusions

The urinary ZEN and DON monitoring system is a valuable tool for identifying contaminated herds and assessing both the short- and long-term impacts of multiple contaminants on herd production and fertility under real field conditions. Implementing this system for herd management could raise awareness among managers and help prevent mycotoxin exposure in cattle herds. Although mycotoxin contamination poses a serious problem in practical herd management, it can be addressed by removing the contaminated feed (as in the ZEN-dominated JB herd) or by reducing its quantity (as in the DON-dominated HF herd). To minimize exposure, we recommend the prompt establishment of novel and accessible measures in production, such as utilization of toxin-binding agents and inactivated feed additives. In conclusion, ZEN-DON contamination adversely affects the productivity, reproductive performance, and health of breeding cattle. Implementing a urinary ZEN-DON monitoring system can be a valuable tool not only for detecting contaminants in cattle production but also for ensuring the safety of food products.

## 5. Materials and Methods

### 5.1. Chemicals and Solvents

ZEN was purchased from MP Biomedicals (Heidelberg, Germany), and metabolites α-ZEL and β-ZEL were purchased from Sigma (St. Louis, MO, USA). Mycotoxin mixtures including DON and 3-Ace-DON were obtained from Romer Labs, Inc. (Newark, NJ, USA). Stock solutions of ZEN, α-ZEL, and β-ZEL (1 μg/mL in methanol) were stored under light protection at 4 °C. High-performance liquid chromatography (HPLC)-grade methanol and acetonitrile were purchased from Fujifilm Wako Pure Chemical Co. (Osaka, Japan) and Sigma-Aldrich, respectively. β-Glucuronidase/arylsulfatase solution was purchased from Merck (Darmstadt, Germany). Sodium acetate was purchased from Kanto Chemical Co., Ltd. (Tokyo, Japan), and Tris was purchased from Nacalai Tesque Inc. (Kyoto, Japan). Oasis HLB Vac Cartridge (6 cc, 2 mg sorbent, 30 μm) was purchased from Waters Co. (Milford, MA, USA).

### 5.2. Identification of Mycotoxin Co-Contaminated Herds as ZEN-Dominant and DON-Dominant Models

#### 5.2.1. JB Breeding Herd as ZEN-Dominant Model

In July 2022, at the request of the managing veterinarian, two JB breeding herds (A and B) in a neighborhood in the Kyushu area of Japan were subjected to urinary ZEN monitoring. The sudden decrease in monthly pregnancy test results in Herd A prompted a request for testing. The urinary ZEN concentration in herd A was high based on our past reported values [9]. As ZEN contamination was strongly suggested in herd A, we informed the managing veterinarian of the test results, recommended the measurement of mycotoxin concentrations in the feed, and requested the collection of urine and blood samples in August. Subsequent measurements of mycotoxin concentrations in the TMR feed diet of herd A confirmed that ZEN contamination exceeded the Japanese standard value (1 mg/kg). However, urinary ZEN monitoring in herd B demonstrated levels of suspected contamination below 1 mg/kg in the feed, as per our previous report [10]. Based on the mycotoxin test results obtained from herd A feed samples, we recommended that the potentially contaminated TMR be discontinued and that the amount of TMR that is discontinued be replaced with other non-contaminated roughages such as rice straw.

#### 5.2.2. HF Breeding Herd as DON-Dominant Model

In July 2022, the managing veterinarian of a dairy farm in the Honshu area of Japan suspected mycotoxin contamination of the cattle feed. This suspicion was based on the observed decline in conception rates, an increase in abortion rates, and an upward trend in mastitis cases. We identified a significant amount of DON in the feed sample, which led the company to initiate discussions on future strategies. Based on the mycotoxin test results obtained from the feed samples, we recommended reducing the amount of potentially contaminated corn silage by half and replacing it with uncontaminated roughage. Blood and urine samples were collected immediately. Our proposal includes both sample collection and continuous monitoring of mycotoxin concentrations using urine and blood samples.

Based on the background described above, these two herds will be used as ZEN (JB herd) and DON (HF herd) contaminated model herds, and after subsequent countermeasures have been taken, urine and blood sampling will be continued, and clinical records will be collected as much as possible.

### 5.3. Dietary Components, Experimental Design, and Sample Collection

The daily feed ratios for each breeding herd are listed in Table 3. In the JB herd, TMR and oat hay were commercially available, whereas WCS and Italian ryegrass silage were homegrown. Concentrate was provided only to cows two months before delivery. Farm HF used two lots of dent corn silage to reduce mycotoxin levels. The contaminated feed was supplemented with fresh uncontaminated feed. The study included two herds with naturally occurring mycotoxin co-contamination, each receiving the same feeding regimen. The JB herd in Kyushu consisted of 29 cows (average age, 3 years and 11 months) with five sample sessions from July to December 2022. The HF herd in Honshu consisted of 30 cows (average age, 3 years and 2 months) with six sampling sessions from August 2022 to May 2024. Additional monitoring will be performed in May 2024. Figure 7 shows the monthly temperature data and schematic of the experimental design. Urine, blood, and roughage samples were collected from the JB (*n* = 3 urine, *n* = 9 blood) and HF (*n* = 10 urine, *n* = 30 blood) herds following procedures [7]. The samples were promptly cooled, transported to the laboratory, and stored at −30 °C after centrifugation. Roughage samples were collected to measure the mycotoxin concentrations. Urine samples from at least two cows were considered sufficient to assess the feed contamination levels [9].

### 5.4. Analytical Methods of Mycotoxins in Feed Samples

In this study, private laboratories conducted measurements of mycotoxin concentrations in feed for each herd. Parallel measurements of mycotoxins (ZEN, DON, aflatoxin B1, ochratoxin A, patulin, fumonisin B1, nivalenol, and sterigmatocystin) were performed using liquid chromatography-tandem mass spectrometry. These measurements followed the multicomponent analysis of mycotoxins as specified in the feed analysis standards of the Food and Agricultural Products Inspection Center, Japan. The screening results for mycotoxin concentrations in feed from the two herds revealed the presence of ZEN and DON at concentrations exceeding the Japanese standard. However, the study excluded other mycotoxin concentrations due to their too low or undetectable levels.

### 5.5. Analytical Methods of ZEN in Urine Samples

Urinary creatinine concentrations were determined using a commercial kit (Sikarikit-S CRE; Kanto Chemical, Tokyo, Japan) and measured using a 7700 Clinical Analyzer (Hitachi High-Tech, Tokyo, Japan). All urine concentrations were expressed as a ratio of creatinine (pg/mg creatinine). ZEN concentrations in the urine were analyzed using an ELISA kit (RIDASCREEN Zearalenon; R-Biopharm AG, Garmstadt, Germany). A urine sample (0.1 mL: 5-fold dilution of the kit) was mixed into 3 mL of 50 mM sodium acetate buffer (pH 4.8) and incubated at 37 °C for 15 h with 10 μL of β-glucuronidase/arylsulfatase solution. The samples were then placed on a C18 solid-phase extraction (SPE) column (Strata; Phenomenex, Torrance, CA, USA) preconditioned with 3 mL of methanol and 2 mL of 20 mM Tris buffer (pH 8.5)/methanol (80:20). The SPE column was washed with 2 mL of 20 mM Tris buffer (pH 8.5)/methanol (80:20) and 3 mL of 40% methanol and then centrifuged for 10 min at 500× *g* to dry. Analytes were eluted gently (15 drops/min) with 1 mL of methanol (80%). From the 1 mL of 80% methanol solution, 0.8 mL was refrigerated for LC-MS/MS measurements, and 0.2 mL was used for ELISA assays. A centrifugal evaporator was used to dry the eluate at 60 °C. To perform the ELISA, the dried residue was redissolved in 50 μL of methanol, 450 μL of sample dilution buffer was added, and the solution was thoroughly mixed. An aliquot of 50 μL was used for the ELISA assay. RIDA SOFT Win (R-Biopharm) was used to determine absorbance at 450 nm using a microplate spectrophotometer. The cross-reactivity rates using this particular ELISA kit for α-ZEL, β-ZEL, and Zeranol were 41.6, 13.8%, and 27.7%, respectively, with a mean recovery rate of 84% ± 14%.

Samples with high ZEN levels were further analyzed using LC-MS/MS to confirm ELISA results and measure ZEN metabolites, α-ZEL and β-ZEL [7,9]. Each urine sample (0.5 mL) was mixed with 3 mL of 50 mM ammonium acetate buffer (pH 4.8) and 8 μL of glucuronidase/arylsulfatase solution and incubated for 12 h at 37 °C. The solution was loaded onto a C18 SPE column, which was preconditioned with 3 mL 100% methanol and 2 mL Tris buffer, followed by the addition of 2 mL Tris buffer and 3 mL 40% methanol. After washing the SPE column with approximately 1 mL 80% methanol, the volume of the eluted solution was adjusted to 1 mL. Then, 5 μL of the reconstituted solution was injected into the LC-MS/MS system (AB Sciex Qtrap 4500 system, Applied Biosystems, Foster City, CA, USA) with the electrospray ionization (ESI) source in the negative mode. The detection limits for ZEN, α-ZEL, and β-ZEL in urine samples were all 0.005 ng/mL, while the mean recovery rates for ZEN, α-ZEL, and β-ZEL were 90%, 109%, and 90%, respectively. Chromatographic separation for ZEN and the metabolites was performed on an InertSustain C18 column (2.1 i.d. × 100 mm, 2 μm; GL Sciences, Tokyo, Japan) at 40 °C using the mobile phase consisting of solvent A (water) and solvent B (methanol). An optimized gradient elution with a flow rate of 200 mL was 50% to 100% B from 0 to 5 min and then held at 100% B for 9 min. The operation of Qtrap was optimized as follows: curtain gas (CUR), 20 psi; ion source (IS) gas 1 and gas 2, 60 and 30 psi; ionspray voltage (IS), −4500 V; temperature (TEM), 500 °C; declustering potential (DP), −125 V for ZEN, −120 V for ZEN metabolites; entrance potential (EP), −10 V; collision energy (CE), −32 V for ZEN, −30 V for metabolites; and collision exit potential (CXP), −9 V for ZEN, −11 V for metabolites. Mass transitions were 317.0 to 175.0 for ZEN and 319.1 to 275.0 for ZEN metabolites.

### 5.6. Analytical Methods of DON in Urine Samples

#### 5.6.1. Preliminary Trial for Urinary DON Measurement by ELISA

The DON concentration in the urine was determined using a commercially available ELISA kit (RIDASCREEN deoxynivalenol; R-Biopharm AG, Darmstadt, Germany). Although the assay kit instructions did not mentione which urine samples were suitable for testing, a preliminary test was conducted to confirm the reliability of the assay for bovine urine samples. The test used mycotoxin mixture solution 4 (Kanto Chemical Co., Inc.) and Biopure Mix 4 (Romer Lab), both containing 10 mg/L 3-Ace-DON, Deoxynivalenol, Nivalenol (NIV), Fusarenon X, HT-2 Toxin, T-2 Toxin, Diacetoxyscirpenol, and Zearalenone. These solutions served as the standard solutions for the recovery test. Briefly, a mixture of the urine samples from 3 different Holstein cows supplemented with 100 μg/L (ppb), 10 μg/L, 1 μg/L, and 0 μg/L of DON (also containing 3-Ace-DON) was used for the recovery tests. Additionally, recovery rates of distilled water supplemented with 100 μg/L (ppb), 10 μg/L, 1 μg/L, and 0 μg/L of DON were also examined. The recovery test was performed twice. The mean recovery rates of DON were ND in the case of supplementation with 100 μg/L (ppb), 542% in the case of supplementation with 10 μg/L, and 585% in the case of supplementation with 1 μg/L. Based on the cross-reactivity data provided by the ELISA kit used in this study, the antibodies showed 100% reactivity with DON, over 100% reactivity with 3-Ace-DON, 19% reactivity with 15-Ace-DON, 4% reactivity with NIV, and less than 1% reactivity with T-2 Toxin. However, different cross-reactivity values were obtained for commercially available DON-ELISA kit antibodies, indicating that the cross-reactivity of 3-Ace-DON in the ELISA kit used in this study was 520 ± 110% and 15-Ace-DON was 3 ± 2% [41]. The results of our preliminary study are consistent with those of a previous report [41]. The ELISA kit proved to be valuable for detection and screening of “DON groups” in bovine urine samples. Subsequently, we established and validated this assay using liquid chromatography–tandem mass spectrometry.

#### 5.6.2. Preliminary Trial for Urinary DON Measurement by LC-MS/MS

For LC-MS/MS analysis, preliminary trials were conducted using direct extraction and pre-incubation with β-glucuronidase and arylsulfatase. The DON was extracted using a commercial immunoaffinity column (DONPREP; R-Biopharm AG, Garmstadt, Germany). After placing a 2 mL urine sample through the column, 10 mL of distilled water was passed through at a rate of 5 mL/minute with air to remove the remaining liquid. The toxin was eluted from the column at 1 drop per second using 100% methanol (1.5 mL) and collected in a glass container. The HLB column was loaded with 0.5 mL of methanol solution and 4.5 mL of Milli-Q. After loading, the column was removed by liquid centrifugation (2500 rpm for 10 min). The target mycotoxins were eluted with 1 mL of methanol/water (4:6, *v*/*v*) and 1 mL of methanol. A glass vial container was used for each elution as described below.

Preliminary trials for extraction methods were conducted to confirm the appropriate quantification of DON and 3-Ace-DON in bovine urine. LC-MS/MS, direct extraction, and pre-incubation with β-glucuronidase/arylsulfatase were conducted. Subsequently, we performed recovery tests. DON recovery rates were 169% with 100 μg/L (ppb), 167% with 10 μg/L, and 105% with 1 μg/L, while 3-Ace-DON recovery rates were 156% with 100 μg/L (ppb), 236% with 10 μg/L, and 462% with 1 μg/L. DON recovery rates in urine samples from 1 Holstein cow supplemented with 100 μg/L (ppb), 10 μg/L, 1 μg/L, and 0 μg/L DON with or without β-glucuronidase/arylsulfatase were 82% and 67% with 100 μg/L (ppb), 88% and 93% with 10 μg/L, and 374% and 87% with 1 μg/L. Sample 1 (1.5 ppb and 1.2 ppb) and sample 2 (0.45 and 0.72 ppb) had urinary DON concentrations with or without β-glucuronidase/arylsulfatase supplementation. In conclusion, the LC-MS/MS system for assessing urine DON concentration is a suitable screening approach, and adding β-glucuronidase or arylsulfatase during extraction does not significantly impact the results.

#### 5.6.3. DON Measurement by LC-MS/MS

The final LC-MS/MS method used an InertSustain C18 column (2.1 i.d., 100 mm, 2 m) in the negative mode to target DON, 3-Ace-DON, and NIV for Qtrap 4500 systems. The detection limit was 0.025 ng/mL in urine when the injection volume was 5 μm. We measured DON, 3-Ace-Don, and NIV under gradient conditions, using a 200-L flow rate. We measured 2% solvent B (methanol) from 0 to 2 min, 2% to 98% B from 2 to 8 min, and maintained 98% B for 12 min. We optimized Qtrap’s operation as follows: The operating pressure was set at 20 psi for CUR, 80 psi for IS gas 1 and 2, 80 psi for gas 2, and 400 °C for TEM. Mass transitions were 295.0 to 265.0 for DON, 337.0 to 307.0 for 3-Ace-DON, and 310.9 to 281.0 for NIV.

### 5.7. Analytical Method of Biochemical Parameters in Serum Samples and Urinary Levels of 8-OHdG

Blood samples (10 mL) from the jugular vein were analyzed for various biochemical parameters using a LaboSpect 7080 autoanalyzer (Hitachi, Tokyo, Japan). Serum vitamin A and E levels were measured via HPLC (Shimadzu, Kyoto, Japan) to assess vitamin depletion. The tests monitored hepatic, renal, and nutritional status, as well as mineral intake. The AMH concentration was measured using a bovine ELISA kit (AnshLabs, Webster, TX, USA) with a detection limit of 11 pg/mL [59]. SAA concentrations were determined using an automated biochemical analyzer (Pentra C200; HORIBA ABX SAS, Montpellier, France) with a specific reagent (VET-SAA ‘Eiken’; Eiken Chemical Co., Ltd., Tokyo, Japan). Serum P4 concentrations were measured via enzyme immunoassay (COSMO FKA302-E, Tokyo, Japan) after diethyl ether extraction, with a detection limit of 0.05 ng/mL [60].

To evaluate oxidative DNA damage, urinary 8-OHdG levels were measured using an ELISA kit (Japan Institute for the Control of Aging, Fukuroi, Japan) with a detection limit of 0.5 ng/mL [61,62]. Diluted urine with distilled water (50 μL: 3 times in HF herd and 20 times in non-contaminated control JB herd) was used for the assay. The absorbance of the samples and standards was measured at 450 nm using a microplate photometer (Multiskan FC; Thermo Fisher Scientific, Tokyo, Japan). 8-OHdG concentrations in the samples were interpolated from a standard curve using log transformation.

### 5.8. Data Management and Statistical Analysis

Correlation analysis quantifies the relationships between biomarkers with the data structure modified using medians to maintain consistent observations [63]. The Kolmogorov–Smirnov test was used to determine the data distribution. For normally distributed data (*p* > 0.05), Pearson’s correlation was used, and for non-normally distributed data (*p* < 0.05), Spearman’s correlation was applied. *p-*values < 0.05, 0.05–0.10, and >0.1 indicated significant, tendency, and non-significant results, respectively. Correlation coefficients (*r*) were categorized as: 0.00–0.19 (very weak), 0.20–0.39 (weak), 0.40–0.59 (moderate), 0.60–0.79 (strong), and 0.80–1.00 (very strong). For ANOVA, normality (Kolmogorov–Smirnov test) and homogeneity (Levene test) were assessed. Post hoc tests included Duncan’s test (homogeneous data) and the Games–Howell test (non-homogeneous data). For non-parametric data, the Kruskal–Wallis test was used, followed by the Mann–Whitney U test for significant results. All analyses were performed using the IBM SPSS Statistics version 25.

## Figures and Tables

**Figure 1 toxins-16-00402-f001:**
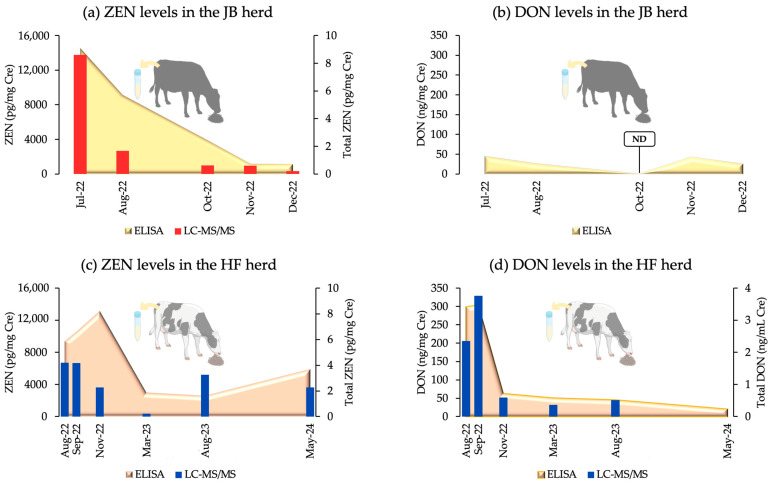
Urinary co-contamination levels in the JB and HF breeding herds, measured by an ELISA and LC-MS/MS. Cre: creatinine; ND: not detected; Total ZEN = ZEN/Cre + α-ZEL/Cre + β-ZEL/Cre; Total DON = DON/Cre + Ac-DON/Cre.

**Figure 2 toxins-16-00402-f002:**
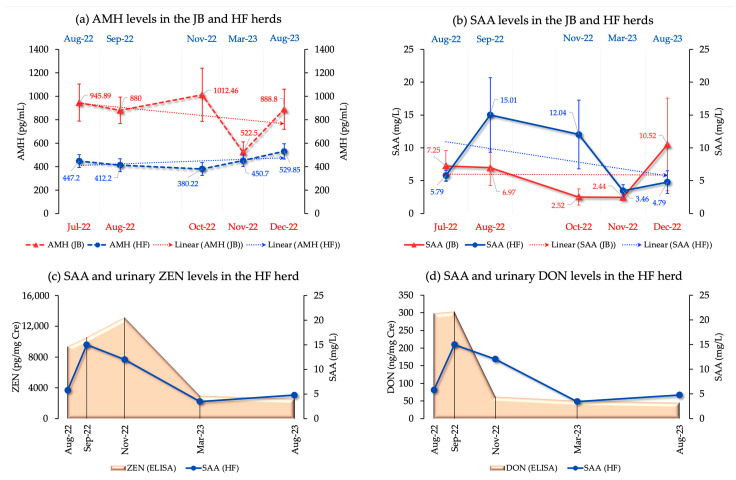
(**a**) AMH levels (trend lines, mean, and SEM) in both JB and HF breeding herds; (**b**) SAA levels (trend lines, mean, and SEM) in both JB and HF breeding herds; (**c**) Changes in both urinary ZEN and SAA levels in the HF breeding herd; (**d**) Changes in both urinary DON and SAA levels in the HF breeding herd.

**Figure 3 toxins-16-00402-f003:**
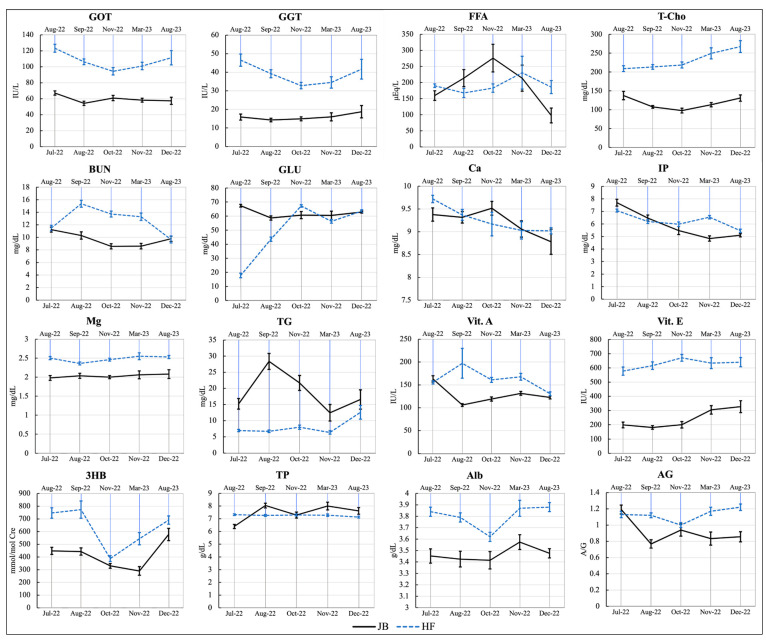
The patterns of blood biochemical analyses (mean ± SEM) of the examined cattle. GOT: glutamate-oxaloacetate transaminase; GGT: gamma-glutamyl transpeptidase; FFA: free fatty acids; T-Cho: total cholesterol; BUN: blood urea nitrogen; GLU: glucose; Ca: calcium; IP: inorganic phosphorus; Mg: magnesium; TG: triglycerides; Vit.A: vitamin A; Vit.E: vitamin E; 3HB: 3-hydroxybutyrate; TP: total protein; Alb: albumin; AG: albumin/globulin ratio.

**Figure 4 toxins-16-00402-f004:**
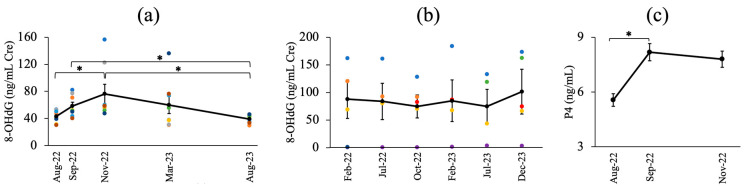
(**a**) The mean value of urinary 8-OHdG/Cre levels in the HF breeding herd (3 times dilution); (**b**) The mean value of urinary 8-OHdG/Cre levels in a non-contaminated breeding herd (20 times dilution); (**c**) Progesterone levels in the HF breeding herd at each sampling period; *: significant differences (*p* < 0.05). Different color dots represent the urinary 8-OHdG/Cre levels in different cows.

**Figure 5 toxins-16-00402-f005:**
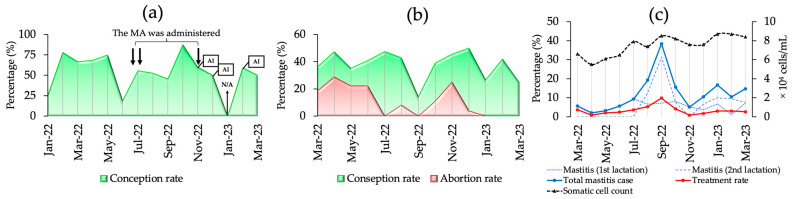
The clinical information of the JB and HF breeding herds. (**a**) The cow conception rate (CCR), feeding modification, and the date of artificial insemination (AI) in the JB breeding herd; (**b**) the CCR and abortion rate in the HF breeding herd; (**c**) the SCC, mastitis cases, and treatment rates in the HF breeding herd; N/A: the data record was not available. MA: mycotoxin adsorbent; ↓↓: the urine indicates a high level of ZEN and DON; ↓: the urine indicates a high level of DON.

**Figure 6 toxins-16-00402-f006:**
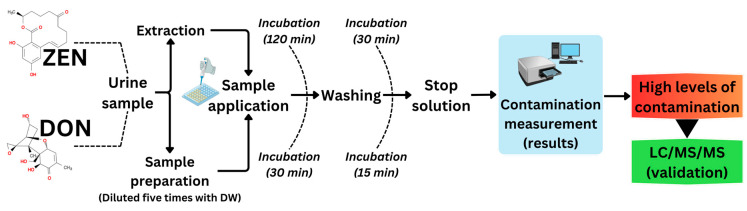
Flow chart of the urinary ZEN-DON monitoring system by ELISA for initial detection and LC-MS/MS for confirmation; DW: distilled water.

**Figure 7 toxins-16-00402-f007:**
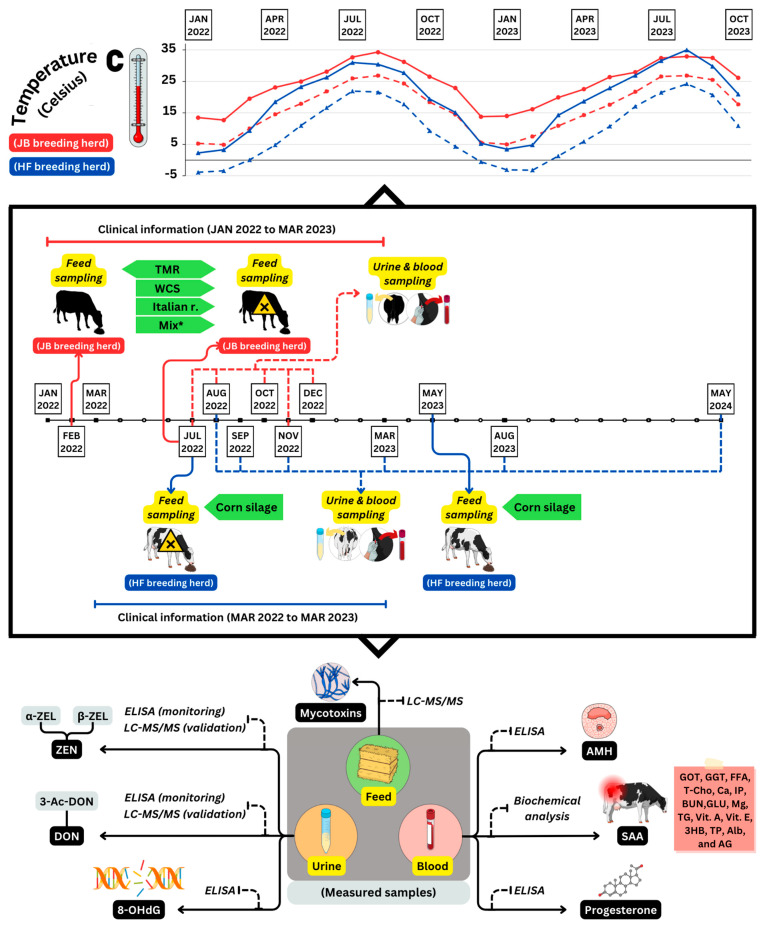
An illustration of the experimental design. The design includes monthly changes of highest and lowest temperatures in degrees Celsius (°C) for the JB and HF breeding herds, as well as comprehensive information on sampling time and the biomarkers analyzed in feed, blood, and urine samples. A harmful mark indicates the first time mycotoxin contamination was detected in feed on both herds (July 2022); TMR: total mixed ratio; WCS: whole crop silage; Italian r.: Italian ryegrass; Mix*: mixture of TMR, WCS, and Italian ryegrass; ZEN: zearalenone; α-ZEL: α-zearalenol; β-ZEL: β-zearalenol; DON: deoxynivalenol; 3-Ac-DON: 3-acetyldeoxynivalenol; AMH: anti-Müllerian hormone; SAA: serum amyloid A; 8-OHdG: 8-hydroxy-2′-deoxyguanosine.

**Table 1 toxins-16-00402-t001:** Results of the JB and HF breeding herd feed sample analyses.

JB Breeding Herd				HF Breeding Herd			
Date	Feed Sample	ZEN (mg/kg)	DON (mg/kg)	Date	Feed Sample	ZEN (mg/kg)	DON (mg/kg)
22 February	Purcased TMR	0.121	ND	22 July	Dent corn silage ^1^	0.1	3
22 July	Purcased TMR	1.791	0.58	22 July	Dent corn silage ^2^ (ori.)	0.062	6.6
22 July	WCS (rice)	0.056	ND	22 July	Dent corn silage ^2^ (dry)	0.145	15.3
22 July	Italian ryegrass (produced in 2020)	0.079	ND	23 May	Dent corn silage	ND	0.09
22 July	Italian ryegrass (produced in 2021)	0.081	ND				
22 July	Mix *	0.733	ND				

* Consists of purchased TMR, WCS rice straw, and Italian ryegrass (produced in 2021) in a feed combination (mix). ^1,2^ The same feed samples were sent to two feed analysis companies. Original sample (ori.), corn silage, had a moisture level of 57.1%. ZEN: zearalenone; DON: deoxynivalenol; ND: not detected.

**Table 2 toxins-16-00402-t002:** The levels of urinary ZEN and DON in JB and HF herds determined through LC-MS/MS.

Herd	Date	*n*	Urinary ZEN (Mean ± SEM)				Urinary DON (Mean ± SEM)		
			ZEN/Cre	α-ZEL/Cre	β-ZEL/Cre	Σ ZEN/Cre	3-Ac-DON/Cre	DON/Cre	Σ DON/Cre
JB	22 July	3	3.66 ± 1.40	1.63 ± 0.68	3.32 ± 1.22	8.61 ± 3.28	-	-	-
	22 August	2	0.12 ± 0.01	0.63 ± 0.07	0.92 ± 0.20	1.66 ± 0.26	-	-	-
	22 October	2	0.12 ± 0.03	0.17 ± 0.02	0.32 ± 0.06	0.62 ± 0.08	-	-	-
	22 November	2	0.13 ± 0.02	0.42 ± 0.16	0.05 ± 0.02	0.59 ± 0.16	-	-	-
	22 December	2	0.04 ± 0.01	0.18 ± 0.05	ND	0.22 ± 0.06	-	-	-
HF	22 August	10	1.57 ± 0.19	0.62 ± 0.06	2.01 ± 0.25	4.20 ± 0.48	0.14 ± 0.03	2.21 ± 0.06	2.35 ± 0.62
	22 September	9	1.67 ± 0.31	0.65 ± 0.13	1.84 ± 0.35	4.16 ± 0.76	0.25 ± 0.04	3.51 ± 0.62	3.76 ± 0.63
	22 November	9	1.03 ± 0.24	1.00 ± 0.19	0.24 ± 0.06	2.27 ± 0.47	0.10 ± 0.01	0.49 ± 0.10	0.59 ± 0.10
	23 March	9	0.13 ± 0.02	0.07 ± 0.02	0.01 ± 0.00	0.22 ± 0.04	0.09 ± 0.02	0.27 ± 0.04	0.36 ± 0.05
	23 August	9	2.69 ± 0.79	0.48 ± 0.04	0.09 ± 0.02	3.26 ± 0.85	0.10 ± 0.02	0.41 ± 0.06	0.51 ± 0.07

JB: Japanese Black; HF: Holstein Friesian; Cre: creatinine; ND: not detected; *-* not evaluated; 3-Ac-DON: 3-acetyldeoxynivalenol; Σ: total value.

**Table 3 toxins-16-00402-t003:** Daily feed ration in the JB and HF breeding herd.

Breeding Herd	Forage Feds	Formula Feeds
Japanese Black (JB)	WCS (rice) [1.5 kg]	TMR [4 kg]
	Italian ryegrass-silage [1.5 kg]	Concentrate * [1.5–2 kg]
	Oats-hay [1 kg]	Calcium-mineral mix [0.04 kg]
Holstein Friesian (HF)	Timothy grass [0.2 kg]	Okara (soybean curd) [7.5 kg]
	Alfalfa grass [1 kg]	Compound feed [7.5 kg]
	Oats-hay [2 kg]	Steam pressed yellow corn [2 kg]
	WCS [7.5 kg]	Wheat bran [1 kg]
	Dent corn-silage (24.5% DM) [8 kg]	Rice bran [2.5 kg]
	Dent corn-silage (27.6% DM) [4.6 kg]	Calcium-mineral mix [0.16 kg]

WCS, whole crop silage; TMR, total mixed ration; Concentrate *: two months before delivery.

## Data Availability

The original contributions presented in the study are included in the article/Appendix A; further inquiries can be directed to the corresponding author.

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
