# Peer review of "Exposure of Cattle Breeding Herds to Naturally Co-Contaminated Zearalenone and Deoxynivalenol: The Relevance of a Urinary Mycotoxin Monitoring System for Herd Health and Food Safety"

_toxins, 2024, doi:10.3390/toxins16090402_

Round 1

Reviewer 1 Report

Comments and Suggestions for Authors

 In the study, the authors perform long-term monitoring and serum biochemical parameters on Cattle after to feed by two herds, Japanese Black (JB) and Holstein Friesian (HF) breeding herds, focusing on AMH levels and acute-phase inflammation. Overall, the manuscript is a comprehensive investigation and well written. The author considered two types of mycotoxins and biochemical parameters from the cattle. In my opinion the results from this research are informative for risk assessment and can be considered for publication in the journal of Toxins. However, a minor modification is required clarified and corrected before publication. They are:

1)    Please, in abstract write the meaning of the abbreviate words: TMR and AMH levels

2)     Section 2. Results.  Line 76: It is indicated that the samples contain aflatoxin (AF) and they were analysed, please can you indicate with which methodology? Also it is important to include each aflatoxin detected, is it AFB1 or AFB2 or AFG1…?

3)     Line 76: Please correct the abbreviation used for ochratoxin (OT), because it is missing to indicate if it is ochratoxin A or B. Also, please indicate the analytical method used to identify this mycotoxin.

4)     Table 1. Please replace Mycotooxins by Mycotoxins.

5)     In table 1 it is indicated values under a number, how it is possible to have this resulps, maybe the authors want to indicate that the value is in the range LD-LQ or bellow LQ, please correct this in the table.

6)     At this point it will be not interesting to include other mycotoxins than ZEN and DON because they are the objective of the study. I suggest to delete all the information regarding others mycotoxins.

7)     Line 202, please replace the type of letter for in vivo and in vitro by cursive. The same in line 335. Please review all the document to correct this.

Author Response

Reviewer 1

Comments and Suggestions for Authors

In the study, the authors perform long-term monitoring and serum biochemical parameters on Cattle after to feed by two herds, Japanese Black (JB) and Holstein Friesian (HF) breeding herds, focusing on AMH levels and acute-phase inflammation. Overall, the manuscript is a comprehensive investigation and well written. The author considered two types of mycotoxins and biochemical parameters from the cattle. In my opinion the results from this research are informative for risk assessment and can be considered for publication in the journal of Toxins. However, a minor modification is required clarified and corrected before publication. They are:

  1. Please, in abstract write the meaning of the abbreviate words: TMR and AMH levels

Accordingly, we have defined the abbreviations total mixed rations (TMR) and anti-Müllerian hormone (AMH) in the abstract. Lines 11 and 15.

  1. Section 2. Results.  Line 76: It is indicated that the samples contain aflatoxin (AF) and they were analysed, please can you indicate with which methodology? Also it is important to include each aflatoxin detected, is it AFB1 or AFB2 or AFG1…?
  2. Line 76: Please correct the abbreviation used for ochratoxin (OT), because it is missing to indicate if it is ochratoxin A or B. Also, please indicate the analytical method used to identify this mycotoxin.

Thank you for the feedback. In the revised version, we have focused on DON and ZEN, and excluded results for other mycotoxins in the dietary feeds. In the Methods section, we outlined the methods used to first screening mycotoxin concentrations in the feeds in accordance with the reviewers feedback on AF, OT, and others as follows: “In this study, private laboratories conducted measurements of mycotoxin concentrations in feed for each herd. Parallel measurements of mycotoxins (ZEN, DON, aflatoxin B1, ochratoxin A, patulin, fumonisin B1, nivalenol, and sterigmatocystin) were performed using liquid chromatography-tandem mass spectrometry. These measurements followed the multicomponent analysis of mycotoxins as specified in the feed analysis standards of the Food and Agricultural Products Inspection Center, Japan. The screening results for mycotoxin concentrations in feed from the two herds revealed the presence of ZEN and DON at concentrations exceeding the Japanese standard. However, the study excluded other mycotoxins concentrations due to their too low or undetectable levels”.  Thank you very much for your understanding. Lines 533-541.

  1. Table 1. Please replaceMycotooxins by Mycotoxins.

We apologize for the spelling mistake. We have corrected the spellings in the revised manuscript.

  1. In table 1 it is indicated values under a number, how it is possible to have this resulps, maybe the authors want to indicate that the value is in the range LD-LQ or bellow LQ, please correct this in the table.

Thank you for your feedback. We have update Table 1 to show “ND” where applicable and have removed all detailed information about mycotoxins other than ZEN and DON from the study.

  1. At this point it will be not interesting to include other mycotoxins than ZEN and DON because they are the objective of the study. I suggest to delete all the information regarding others mycotoxins.

Thank you for this insightful suggestion. Accordingly, we deleted all detailed information on mycotoxins other than ZEN and DON in the revised manuscript.

  1. Line 202, please replace the type of letter for in vivo and in vitro by cursive. The same in line 335. Please review all the document to correct this.

We apologize for incorrect presentation. We have corrected the issue by italicizing in vivo and in vitro throughout the manuscript.

Reviewer 2 Report

Comments and Suggestions for Authors

 The study presents comprehensive data, but the results section would benefit from clearer highlighting of the key findings related to mycotoxin contamination and its impact on herd health.

The methodology is well-detailed, but additional clarity on the statistical analysis used to correlate mycotoxin levels with health parameters would strengthen the manuscript.

While the discussion provides useful insights, it could be expanded to include comparisons with other studies on the impact of mycotoxin contamination on cattle fertility and productivity.

The figures and tables are informative, but they should be more directly referenced in the text to ensure readers can easily follow the data presented.

Comments on the Quality of English Language

No

Author Response

Reviewer 2

Comments and Suggestions for Authors

The study presents comprehensive data, but the results section would benefit from clearer highlighting of the key findings related to mycotoxin contamination and its impact on herd health.

  1. The methodology is well-detailed, but additional clarity on the statistical analysis used to correlate mycotoxin levels with health parameters would strengthen the manuscript.

Thank you for this insightful comment. We agree with your comment. However, as stated within the discussion section, the clinical manifestations depend on the specific trichothecene, dose, method of exposure, and type of exposure, suggesting that natural and experimental exposures and field contaminated materials may have different effects than those of pure toxins. Therefore, one of the objectives of this study was to develop a database for exploring the relationship between in vivo mycotoxin contamination levels and clinical symptoms. We believe that further research is needed to address the important and valid points raised by the reviewer. Thank you for your understanding.

  1. While the discussion provides useful insights, it could be expanded to include comparisons with other studies on the impact of mycotoxin contamination on cattle fertility and productivity.

Thank you for your thoughtful comment. This field study was based on novel concepts and methods used in cattle herds. To the best of our knowledge, there are no other reports on in vivo ZEN and DON exposure levels (urinary levels), fertility, and productivity in cattle other than those published by our research group previously; therefore, it is difficult to compare our present results with previous reports. Based on the novel and significant results observed in the present study, future research involving larger sample sizes is essential to validate these findings. Thank you for your understanding. 

  1. The figures and tables are informative, but they should be more directly referenced in the text to ensure readers can easily follow the data presented.

We apologize for the confusion. Due to the large number of figures and tables included, we intended to provide a direct description of each figure and table pertaining to our results. We are unsure to what you are referring to. If you could please clarify this, we will ensure a direct explanation is added during the editorial stage. We appreciate to your remark and thank you for your understanding.

Reviewer 3 Report

Comments and Suggestions for Authors

This word used the herds as models of ZEN- and DON-dominant contamination to evaluate the toxic effects on health and fertility. The result showed ZEN-DON contamination adversely affects breeding cattle's productivity and ZEN-DON monitoring system can be a valuable tool for detecting contaminants.

1.       Line469 what does substantial contamination levels mean? Contamination percentage or concentration over 1 ppm?

2.       Section 2.1 ppm should be revised by mg/kg for the contamination concentration of mycotoxins.

3.       Line90 “ZEN levels peaked in July 2022 and declined until December 2022”. Does the mycotoxin level changed due to the weather or the change of feeds?

4.       The title of this work mentioned this work aims to investigate the relevance of mycotoxins exposure for herd health and food safety as well. However, there are limited information about food safety.

Author Response

Reviewer 3

Comments and Suggestions for Authors

This word used the herds as models of ZEN- and DON-dominant contamination to evaluate the toxic effects on health and fertility. The result showed ZEN-DON contamination adversely affects breeding cattle's productivity and ZEN-DON monitoring system can be a valuable tool for detecting contaminants.

  1. Line469 what does substantial contamination levels mean? Contamination percentage or concentration over 1 ppm?

Thank you for this remark, and we apologize for the confusion. According to the remark, we revised the sentence as follows: “However, urinary ZEN monitoring in herd B demonstrated levels of suspected contamination below 1 ppm in the feed, as per our previous report [10]Lines  481-482.

  1. Section 2.1 ppm should be revised by mg/kg for the contamination concentration of mycotoxins.

Accordingly, we have revised “ppm” to “mg/kg” in all of Section 2.1.

  1. Line90 “ZEN levels peaked in July 2022 and declined until December 2022”. Does the mycotoxin level changed due to the weather or the change of feeds?

Thank you very much for your kind remarks, and we apologize for the confusion caused by the lack of an explanation regarding the gradual decrease in urinary ZEN levels. The previous manuscript did not adequately describe the countermeasures implemented after the detection of ZEN contamination in the feed in July 2022. Accordingly, we have revised the manuscript as follows; “Based on the mycotoxin test results obtained from the herd A feed samples, we recommended that the potentially contaminated TMR be discontinued and that the amount of TMR that is discontinued be replaced with other non-contaminated roughage such as rice straw”. Lines   482-485.

  1. The title of this work mentioned this work aims to investigate the relevance of mycotoxins exposure for herd health and food safety as well. However, there are limited information about food safety.

Thank you very much for your perceptive remarks. We agree with the reviewer’s comment. However, in this study, beef and dairy cattle herds were monitored. Further, this study assessed the dynamics of mycotoxin infiltration at a dairy meat production site, which is a major source of human food. We believe that food safety begins with not only mycotoxin contamination of livestock products but also with the health of the herd at the source of the contamination. In line with the reviewer’s feedback, we have revised the introduction section as follows; “We monitored beef and dairy cattle herds to examine the dynamics of mycotoxin contamination at meat and milk production sites, which are major sources of human food. Furthermore, food safety depends not only on mycotoxin contamination of livestock products but also on the health of cattle at the source of contamination”. Lines 44-47.

Round 2

Reviewer 3 Report

Comments and Suggestions for Authors

Thank you for the response and revision. All the comments have been revised and I think this work could be accepted in the current version.